# Very Stable Two Mega Dalton High-Molecular-Mass Multiprotein Complex from Sea Cucumber *Eupentacta fraudatrix*

**DOI:** 10.3390/molecules26185703

**Published:** 2021-09-21

**Authors:** Anna M. Timofeeva, Irina A. Kostrikina, Pavel S. Dmitrenok, Svetlana E. Soboleva, Georgy A. Nevinsky

**Affiliations:** 1Institute of Chemical Biology and Fundamental Medicine, Siberian Division of Russian Academy of Sciences, Lavrentiev Ave. 8, 630090 Novosibirsk, Russia; bezukaf@mail.ru (A.M.T.); irina@kostrikina.ru (I.A.K.); sb543@ngs.ru (S.E.S.); 2G. B. Elyakov Pacific Institute of Bioorganic Chemistry, Far Eastern Brunch of the Russian Academy of Sciences, 159 Pr. 100 Let Vladivostoku, 690022 Vladivostok, Russia; paveldmt@piboc.dvo.ru

**Keywords:** model of tissues and body regeneration, sea cucumber *Eupentacta fraudatrix*, very stable 2000 kDa protein complex

## Abstract

In contrast to many human organs, only the human liver can self-regenerate, to some degree. Some marine echinoderms are convenient objects for studying the processes of regenerations of organs and tissues. For example, sea cucumbers *Eupentacta fraudatrix* can completely restore within several weeks, the internal organs and the whole body after their division into two or three parts. Therefore, these cucumbers are a very convenient model for studying the general mechanisms of regeneration. However, there is no literature data yet on which biomolecules of these cucumbers can stimulate the regeneration of organs and the whole-body processes. Studying the mechanisms of restoration is very important for modern biology and medicine, since it can help researchers to understand which proteins, enzymes, hormones, or possible complexes can play an essential role in regeneration. This work is the first to analyze the possible content of very stable protein complexes in sea cucumbers *Eupentacta fraudatrix*. It has been shown that their organisms contain a very stable multiprotein complex of about 2000 kDa. This complex contains 15 proteins with molecular masses (MMs) >10 kDa and 21 small proteins and peptides with MMs 2.0–8.6 kDa. It is effectively destroyed only in the presence of 3.0 M MgCl_2_ and, to a lesser extent, 3.0 M NaCl, while the best dissociation occurs in the presence of 8.0 M urea + 0.1 M EDTA. Our data indicate that forming a very stable proteins complex occurs due to the combination of bridges formed by metal ions, electrostatic contacts, and hydrogen bonds.

## 1. Introduction

It is imperative to study many socially significant diseases associated with the dysfunction of various organs. Some organs of humans, such as the liver, are to some extent capable of self-regeneration. However, some other human organs are wholly or almost incapable of self-regeneration. The ability to restore lost and damaged structures of various organs is the most important adaptation mechanism in some marine organisms [1,2,3,4,5]. Marine echinoderms, including sea cucumbers, are very suitable objects for studying the processes of restoration-regeneration of organs and tissues. Many of them can make relatively rapid and complete restoration of lost body parts after a wide variety of injuries [1,2,3,4,5]. For example, many sea cucumbers can completely restore the internal organs lost after evisceration or their scission into two or three parts; they completely restore the whole body within several weeks. However, the mechanisms of their regeneration have not yet been studied, and it is not clear what components of these organisms (proteins, enzymes, hormones, other compounds, or their possible complexes) can trigger regeneration mechanisms and are necessary for the complete regeneration of their organs and tissues. Therefore, identifying the components of sea cucumbers that may be important for regeneration processes is an important step in understanding regeneration mechanisms.

It was proposed that a very large number of biological processes are implemented due to different protein complexes [6]. Many cellular processes depend on the functioning of several enzymes, which are often associated with each other forming stable or temporary protein complexes to increase the efficiency, specificity, and speed of metabolic pathways [7]. The association of various enzymes and proteins can lead to the formation of polyfunctional molecules and the expansion of their biological functions.

Human protein complexes associated with placental membranes were recently analyzed using SDS-PAGE and MALDI mass spectrometry [8]. Overall, 733 unique proteins, including 34 protein complexes, were identified. The possibility of the existence of multi-protein complexes in the soluble fraction of a homogenate of human milk [9], placenta [10,11], and eggs of sea urchins [12] was recently analyzed. The extremely stable complexes of different proteins (≈1000 ± 100 kDa) were isolated from milk, placentas, and eggs of sea urchins by FPLC gel filtration [9,10,11,12]. These three protein complexes were stable in the presence of MgCl_2_ and NaCl in high concentrations. All complexes were dissociated only in the presence of 8 M urea + 1.0–3.0 M NaCl, or MgCl_2_ [9,10,11,12]. In the case of the placentas, milk, and eggs of sea urchins, the stable complexes, according to SDS-PAGE, contain several, yet different proteins having molecular masses (MMs) from 14 to 79.3 kDa [9,10,11,12]. It was shown that placenta SPCs contain twelve proteins: hemoglobin, alkaline phosphatase, cytoplasmic actin, human serum albumin, chorionic somatomammotropin hormone, heat shock protein beta-1, peroxiredoxin-1, 78 kDa glucose-regulated protein, protein disulfide isomerase A3, serotransferrin, annexin A5, and IgGs [10,11]. Human milk SPCs contain α-lactalbumin and lactoferrin as major proteins, whereas human milk albumin, alkaline phosphatase, β-casein, sIgAs, and IgGs were present in moderate amounts [9]. Identification of sea urchin complex proteins was performed using several different databases, including databases specific to sea invertebrates [12]. Major proteins in sea urchin complexes exhibited homology with keratin (type II cytoskeletal 1; Homo sapiens), alkaline phosphatase of *Pseudomonas fluorescens*, human lactotransferrin, glucose-6-phosphatase of *Strongylocentrotus purpuratus*, and one protein similar to *Alveinella pompejana* (cDNA clone CAGA18424 5′, mRNA sequence). Several remaining major and average proteins were impossible to identify even through searches for their possible homology with proteins of other different organisms [12]. The only common protein in all three stable complexes was alkaline phosphatase. This suggests that stable complexes of different protein compositions can also exist in other organisms and have a specific biological role.

We proposed that such very stable protein complexes can also be in sea cucumber *Eupentacta fraudatrix.* In contrast to individual enzymes and proteins and other biological components, their complexes can usually have other bodily functions. Identification and characterization of proteins and their different complexes seem to be very important for understanding regeneration mechanisms.

In this study, we used different methods (gel filtration, SDS-PAGE analysis, light scattering, MALDI mass spectrometry) to analyze, for the first time, whether any stable protein complexes can exist in sea cucumber *Eupentacta fraudatrix*.

## 2. Results

### 2.1. Isolation and Analysis of Sea Cucumbers Protein Complex

Seven samples of sea cucumbers were subjected to homogenization and used for the study. Homogenates were concentrated and subjected to FPLC gel filtration on the Sepharose 4B. Sepharose 4B efficiently separates proteins with molecular masses (MMs) of 60–20,000 kDa. A typical profile of gel filtration is shown in Figure 1A.

It is apparent that one protein peak with a molecular mass comparable with that for Blue dextran (≈2000 kDa) is separated from other various proteins. After gel filtration of non-concentrated extracts of the cucumbers, we revealed the same protein peak of high molecular mass, ≈2000 kDa. This means that this specific multiprotein complex exists in non-concentrated extracts of cucumbers. Repeated gel filtration of the purified complex (Figure 1A) on the Sepharose 4B showed only one peak with the same MM, ≈2000 kDa (Figure 1B). The only treatment of the complex under very harsh conditions (50 mM acidic buffer (pH 2.6), 8.0 M urea, 2.0 mM DTT, and 3.0 M NaCl) led to its partial dissociation into separate components (Figure 1C). The top of the principal peak corresponding to the complex shifted slightly compared to the peak before the complex treatment, but peaks of proteins with lower molecular weights appeared.

### 2.2. Light Scattering Assay

For a more detailed analysis of the stability of the complex, we used the light scattering (LS) method. Urea, MgCl_2_, and NaCl at high concentrations usually efficiently dissociate noncovalent complexes of various proteins, including immunocomplexes. It was earlier shown that buffer containing 8.0 M urea best destroys very stable complexes from human milk, placentas and sea urchin eggs [9,10,11,12].

According to the LS data, the complex from cucumbers was stable in a buffer containing 0.15 M NaCl (Figure 2). Quite unexpectedly, the treatment of the complex with a buffer containing 8.0 M urea led to a relatively small decrease in LS by only 20.0 ± 2.0% (Figure 2) compared to 30–70% in the case of stable complexes from human milk, placentas, and sea urchin eggs [9,10,11,12].

Incubation of the complex in the presence of 1.0 M NaCl leads to a decrease in LS by 23.7 ± 3.0%, while 1.0 M MgCl_2_ decreased LS by 31.8 ± 3.0%. However, in the presence of 3.0 M NaCl and MgCl_2_, the LS was reduced by 45.0 ± 3.0% and 65.0 ± 4.0%, respectively. Interestingly, a more effective decrease in LS was observed in the presence of 8.0 M urea together with 1.0 M MgCl_2_ sodium (55.2 ± 3.0%) in comparison with urea and magnesium salt alone. However, unexpectedly, the addition of 2.0 mM DTT to a mixture of 8.0 M and 1.0 M MgCl_2_ barely affected the dissociation of the complex (56.8 ± 3.0%). This may indicate in favor of the fact that in contrast to the complexes from human milk, placenta, and sea urchin [9,10,11,12], disulfide bridges (S-S) between protein molecules in the sea cucumber complex may play a lesser role in the complex formation. Thus, it is apparent that ≈2000 kDa protein complex from the soluble fraction of sea cucumber is very stable. However, in the presence of 3.0 M NaCl and MgCl_2_, the LS was decreased by 45.0 ± 3.0% and 65.0 ± 4.0%, respectively. A somewhat unexpected situation appeared when a decrease in LS was observed for 8.0 M urea (20.0 ± 2.0%) and 0.1 EDTA (24.0 ± 2.0%) separately and in their joint presence (75.2 ± 5.0%) (Figure 2). It is possible that the rupture of some hydrogen bonds between the proteins leads to an increase in the availability of EDTA to remove metal ions that form bridges in the inner globule of the complex. At the same time, the opposite situation is not excluded when the removal of metal ions with EDTA allows urea to destroy a larger number of hydrogen bonds. In other words, EDTA and urea have a synergistic effect during the destruction of a stable complex.

Thus, it should be assumed that the bridges with metal ions formed between complex proteins may be very important for the formation and stability of the sea cucumber complex. The totality of the data obtained indicates that the formation of a very stable complex of cucumber proteins occurs due to combination of bridges formed by metal ions, electrostatic contacts, and hydrogen bonds.

### 2.3. SDS-PAGE Analysis of the Complex Proteins

Using SDS-PAGE, we analyzed proteins of the complex (Figure 3).

Fifteen reliably tested proteins with different MMs were found. One protein was especially major (85.2 ± 5.0 kDa). The protein with MM 17.3 ± 1.5 kDa) can also be classified as major. The rest of the proteins were contained in the complex in significantly smaller amounts and were moderate or minor. After SDS-PAGE, all 15 proteins were subjected to standard trypsinolysis in order to identify those using MS and MS/MS MALDI mass spectra. For most protein hydrolyzed, good MS and MS/MS spectra were obtained. However, it was impossible to identify any of them. This is due to the absence of data on such cucumber proteins in the databases of sea invertebrates. Thus, currently, protein MMs could be estimated only with the help of SDS-PAGE.

### 2.4. Peptides of the Complex 

One of the specific features of stable complexes from human milk and placenta, as well as sea urchin eggs, is that, in addition to proteins with a molecular weight of >10 kDa, they contain a large number of small proteins and peptides with MMs of 2–9 kDa [9,10,11,12]. All small proteins and peptides <10 kDa are soluble in acetic acid, and during gels staining with Coomassie, they are usually washed out of the gel. To analyze the possible content of peptides in the sea cucumber complex, we subjected the complex to destruction using concentrated trifluoroacetic acid in the dioxane as in [9,10,11,12]. Then, this extract was filtered through the Amicons filter, passing compounds less than 10 kDa (to remove large proteins >10 kDa). The resulting fraction of small proteins and peptides was analyzed using MALDI mass spectrometry. Figure 4 shows the spectrum of the components of this fraction.

Based on the analysis of seven spectra, it was found that this fraction contains about 21 reliably tested small peptides and proteins with MMs from 2.0 to 8.6 kDa. When using an α-cyano-4-hydroxycinnamic acid matrix, peptides and small proteins can be more reliably identified than any other compounds and large proteins [9,10,11,12]. Peaks of oligonucleotides, lipids, sugars, and some other compounds could appear in MALDI spectra only if their content in mixtures analyzed is 100–1000-fold higher than that of small proteins and peptides. Therefore, it should be assumed that, as in the case of protein complexes from the placenta, milk, and sea urchins [9,10,11,12], all peaks of the spectra (Figure 4) most likely correspond to different peptides and small proteins. Nevertheless, the fraction <10 kDa was subjected to treatment with proteinase K. After the incubation, all peaks with MMs higher >3 kDa disappeared, and smaller hydrolysis products were formed; MMs of smaller products do not correspond with those of the initial mixture.

## 3. Discussion

Reiterating the above information, human milk and placenta and sea urchin eggs contain very stable protein complexes with a molecular weight of 1000 ± 100 kDa [9,10,11,12]. Considering this, it was completely unexpected to find in the extracts of sea cucumber a very stable complex with an MM comparable to that of blue dextran—2000 kDa (Figure 1). The very high stability of this complex indicates that its formation cannot result from a random association of sea cucumber proteins. A feature of this protein associate compared with the previously described stable complexes [9,10,11,12] is that it is relatively weakly degraded in the presence of only 8.0 M urea. At the same time, the efficiency of its dissociation when treated with 1–3 M NaCl and MgCl_2_ in high concentrations is much more efficient (Figure 2). It is known that urea destroys mainly hydrogen bonds between protein molecules, while salts in high concentration primarily disrupt electrostatic contacts. It is possible that in the case of a sea cucumber stable complex, the proteins and peptides included in its composition form, to a lesser degree than hydrogen bonds with each other than electrostatic contacts. Nevertheless, the incubation of the complex with the mixtures of 8.0 M urea and 1.0 M MgCl_2_ leads to an acceleration of complex decomposition compared to that in the presence of only one of these components (Figure 2). The decrease in LS for 8.0 M urea (20.0 ± 2.0%) and 0.1 M EDTA (24.0 ± 2.0%) separately is significantly lower than with their joint presence (75.2 ± 5.0%); EDTA and urea have a synergistic effect during the destruction of a stable complex.

Thus, metal ions can be important for forming contacts between the molecules of proteins and peptides that make up the complex. Consequently, in general, forming a very stable complex occurs with the formation of a combination of various kinds of contacts, including hydrogen bonds, electrostatic interactions, and contacts of proteins due to metal ions.

At this stage, it was possible to detect (using SDS-PAGE) only 15 proteins, some of which are major, middle, and minor. High-quality MS and MS/MS spectra were obtained for tryptic hydrolysates of all proteins detected by SDS-PAGE. However, a database of sea cucumber proteins does not exist, and using various other databases, we were unable to indicate proteins corresponding to proteins bands after SDS-PAGE.

A fraction of small proteins and peptides was obtained by filtering the complex destroyed by trifluoroacetic acid in dioxane through filters that pass molecules with a molecular weight of <10 kDa. Analysis of the components of this fraction showed that the complex contains about 21 different small proteins and peptides. In this case, five components with MMs 2147.5, 3472.6, 3978.6, 6023.2, and 6950.5 Da (Figure 4) can be attributed to the major components and the rest to the middle or minor peptides of the protein complex. It is possible that these peptides can play an important role both in the formation of a stable complex and in its stabilization. As can be seen from the data on the gel filtration of the complex after its processing under severe conditions, there still remains a small crustal structure of the complex with slightly lower MMs than that of the initial complex (Figure 1C). Therefore, it is possible that not all peptides were recovered from the complex after treatment with trifluoroacetic acid in dioxane.

## 4. Materials and Methods

### 4.1. Materials

Most chemicals used for this study were from Sigma (St. Louis, MO, USA). Sepharose 4B (GE Healthcare Life Sciences, New York, USA). Sea cucumber *Eupentacta fraudatrix* were collected from Peter the Great Bay, Sea of Japan. Samples of cucumbers were frozen to −40 °C and stored until the preparation of the extracts.

### 4.2. Sea cucumber Extracts Preparation 

Equal parts (10.0 g) of seven sea cucumbers (after bowel removal) were used to obtain the supernatant. All slices of cucumbers before lysis were carefully treated 3 times with a buffer containing antibiotics (10,000 U Pen/mL penicillin, 10,000 ug Strep/mL ampicillin, 25 ug Amphotericin B/mL streptomycin) to inactivate and remove possible bacteria on their surface. Then pieces of sea cucumber were homogenized in a buffer containing 10 mM Tris-HCl, pH 8.0; 1 M NaCl, 1 mM DTT, 1mM EDTA, and the above antibiotics, in a volume ratio of 1:10. The sea cucumber homogenate was centrifuged 16,500× *g* for 40 min 2 times, then filtered through a 0.2 nm filter, dialyzed 3 times against water, and concentrated using a dialysis bag in an airstream at 4 °C. The supernatant was again centrifuged at 12,000× *g* 4 °C for 10 min and subjected to FPLC gel filtration on Sepharose 4B.

### 4.3. Purification of Stable Complexes by Gel Filtration 

The purification of the complex was implemented using Sepharose 4B as in [9,10,11,12]. The concentrated homogenate (0.5–1.0 mL) was applied on Sepharose 4B a column with (volume 50 mL) equilibrated in TBS (20 mM Tris HCl pH 7.5; 0.15 M NaCl) and GE Akta Purifier chromatograph (Chicago, IL, USA). All fractions eluted (1 mL) were collected. To analyze the cucumber complex stability, it was preliminarily incubated for 30 min 50 mM acidic Tris-Gly (pH 2.6) buffer containing 8 M urea, 2 mM DTT, 3.0 M NaCl, at 37 °C. Then the complex was applied to the Sepharose 4B column, as described above. The complex and proteins were monitored by absorbance at 280 nm (A_280_). All procedures were performed under sterile conditions.

### 4.4. SDS-PAGE Assay 

SDS-PAGE analysis of complex proteins was performed in a 4–17 % gradient gel containing 0.1% SDS, according to Laemmli. Prior, SDS-PAGE preparations of proteins (20–45 μg) were incubated using a buffer A containing 50 mM Tris-HCl (pH 6.8), 10% glycerol, 1% SDS, 0.025% bromophenol blue, 10 mM EDTA for 7 min at 100 °C and then applied to the gel. Proteins were stained with Coomassie R-250 or colloid silver. Analysis of tryptic hydrolysates of proteins after their SDS-PAGE was performed using MS and MS/MS data of MALDI-TOF mass spectrometric analysis as in [9,10,11,12].

### 4.5. Analysis of Peptides and Small Proteins

The destruction of stable complex was carried out by analogy with the destruction of different cells according to [13,14,15]. To 10 μL of purified complex, 40 μL of trifluoroacetic acid (TFA) was added; the mixture was shaken for 30 min. After the addition of 150 μL of H_2_O the mixture was shaken for 10 min. After adding acetonitrile (200 μL) and shaking the mixture for 12 min, it was centrifuged for 25 min at 15 rpm in an Eppendorf centrifuge to remove the insoluble precipitate. Peptides and small proteins (<10 kDa) were obtained using Amicon membranes transmitting proteins with MMs <10 kDa. To improve the MALDI mass spectra, the solutions obtained were concentrated to a volume of 10 μL, and their compounds were purified from salts using ZIPTip Pipette Tips C18 (Sigma-Aldrich) according to a standard procedure. The peptides were eluted from Tips with 85% acetonitrile. The resulting preparations were analyzed by MALDI mass spectrometry, as described below.

To control the solution 0.01 mg/mL proteinase K was added. After incubation of the mixtures at 30 °C for 10 h, 2 μL of the solution was used to analyze MMs by MALDI mass spectrometry, as described below.

### 4.6. MALDI Mass Spectrometry Analysis of Proteins

For analysis of peptides and tryptic hydrolyzates of proteins after SDS-PAGE, MALDI mass spectrometry was used. To 1.2 μL of all peptide preparations analyzed 1.2 μL of a saturated solution of α-cyano-4-hydroxycinnamic acid solved in 0.1% acetonitrile and trifluoroacetic acid (1:2) was added, and 1.2 μL of the mixture obtained was applied on the MALDI steel plate, air-dried, and used for the analysis. The analysis of peptides was carried out using the Reflex III system from Bruker Company (Frankfurt, Germany): 337-nm nitrogen laser VSL-337 ND, 3 ns pulse duration. All MALDI spectra were calibrated using standard protein mixtures II and I (Bruker Daltonic, Bremen, Germany) in the internal /or external calibration mode. The analysis of peptide MMs after proteins standard hydrolysis with trypsin was performed using Protein Calculator v3.3 (Scripps Research Institute; La Jolla, CA, USA).

## 5. Conclusions

In this study, for the first time, researchers revealed a very large and very stable peptide-protein complex of sea cucumbers *Eupentacta fraudatrix*. This complex is very large (2000 kDa). It is effectively destroyed only in the presence of 3.0 M NaCl or MgCl_2_, but maximum dissociation is observed in the joint presence of 8.0 M urea and 0.1 M EDTA. Overall, the data obtained indicate that forming a very stable complex of sea cucumber proteins occurs due to the sum of bridges formed by metal ions, electrostatic contacts, and hydrogen bonds.

## Figures and Tables

**Figure 1 molecules-26-05703-f001:**
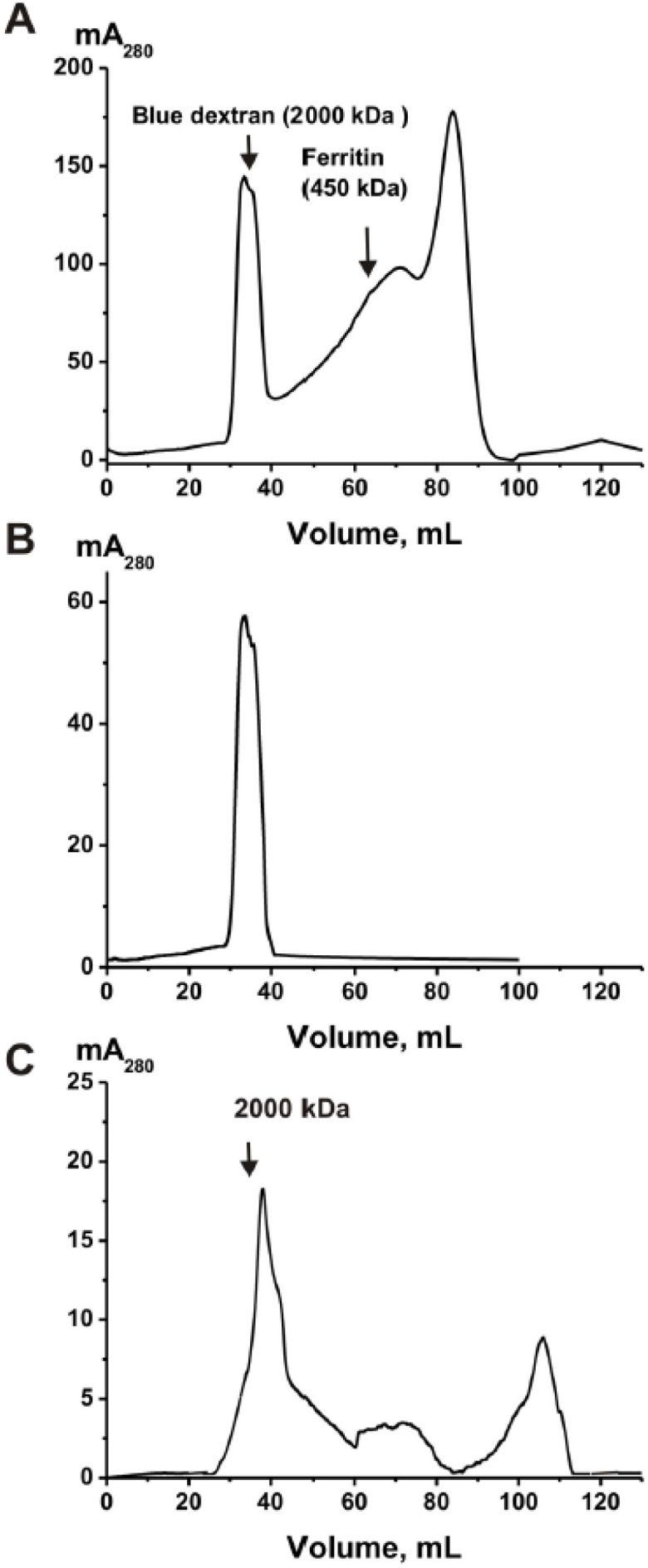
Isolation and analysis of sea cucumber complex. FPLC gel filtration of sea cucumber homogenate proteins on a Sepharose 4B column (**A**). Gel filtration of the purified complex (**A**) on the Sepharose 4B, MM = ≈2000 kDa (**B**). Gel filtration of the complex after its treatment using harsh conditions: 50 mM acidic buffer (pH 2.6), 8.0 M urea, 2 mM DTT, and 1.0 M NaCl (**C**). In all Panels, (^__^)—absorbance at 280 nm (A_280_).

**Figure 2 molecules-26-05703-f002:**
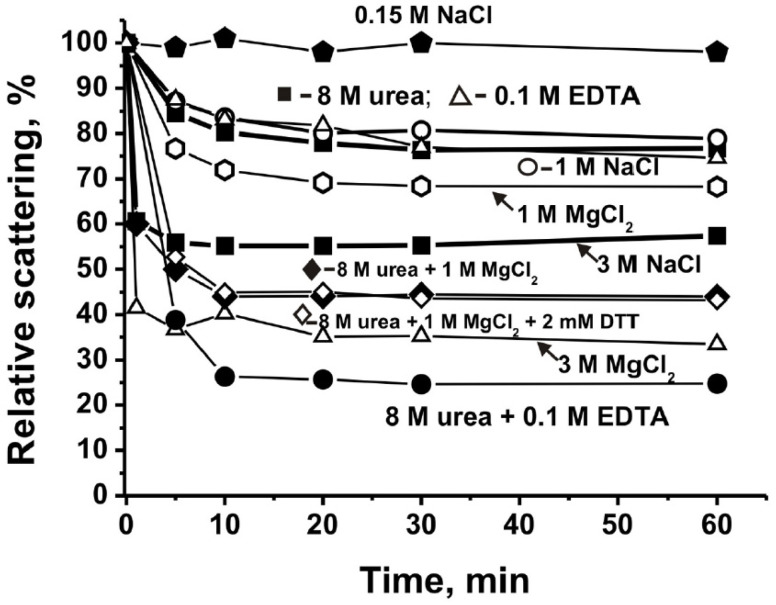
Typical examples of the time changes in the light scattering (LS) of the cucumber complex (0.005 mg/mL) in the presence of 20 mM Tris-HCl buffer (pH 7.5) containing urea, NaCl, MgCl_2_, DTT, and EDTA in various concentrations and different combinations, detailed in the Figure.

**Figure 3 molecules-26-05703-f003:**
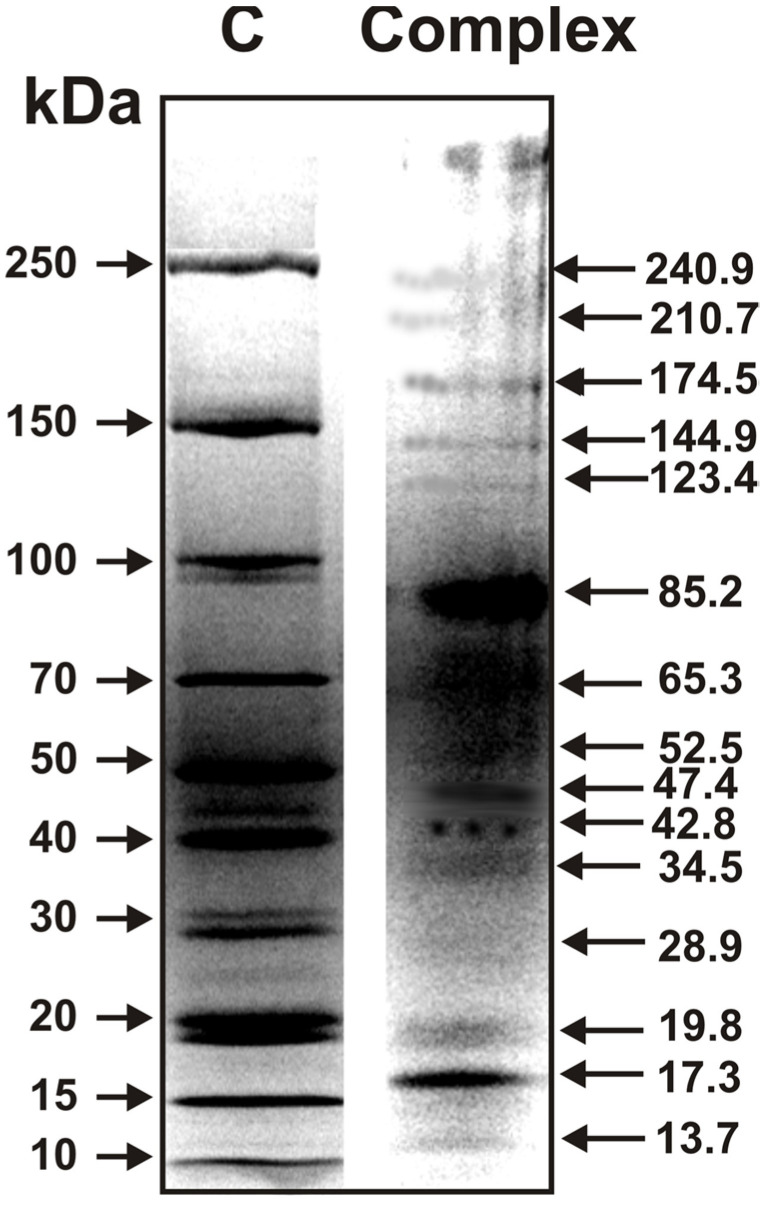
SDS-PAGE analysis of the proteins of sea cucumber complex (25 μg) using 3.0–17.0% gradient gel. The arrows of Lane C indicate the positions of markers of molecular masses. See Materials and Methods for other details.

**Figure 4 molecules-26-05703-f004:**
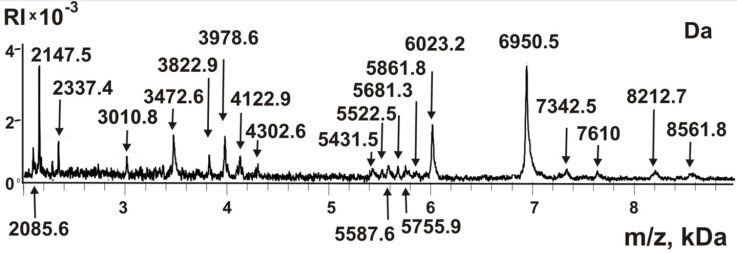
MALDI spectra of peptides and small proteins of sea cucumber complex. The fraction of peptides and small proteins was prepared by dissociation of the complex by trifluoroacetic acid in dioxane and its filtration through filters passing compounds with a MMs s <10 kDa. See Materials and Methods for other details.

## Data Availability

All data is given in this article.

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
