# Peer review of "Very Stable Two Mega Dalton High-Molecular-Mass Multiprotein Complex from Sea Cucumber Eupentacta fraudatrix"

_molecules, 2021, doi:10.3390/molecules26185703_

Round 1

Reviewer 1 Report

The manuscript is focused on the identification of protein complex from sea cucumber (Eupentacta fraudatrix). The complex of MW 2 MDa was isolated by GPC and further analyzed by gel electrophoresis and MALDI. In total 15 proteins and 21 peptides were identified with those methods. A shortcoming of this manuscript is poor characterization of the constituents and qualitative analysis of the complex. In the case of proteins of MW >10 kDa only SDS-PAGE results, while for peptides only MALDI results were presented. No other quantitative or qualitative experiments were attempted to characterize the constituents of the complex. This part of the manuscript should be extended. As a minor remarks the presentations of uncertainties should be corrected. The authors indicated single or double digit uncertainty while corresponding results were presented as if they were measured with lower or higher accuracy. For example 55.2+/-3 or 24 +/- 2.0% (section 2.2 and 2.3). Instead, single digit uncertainty and only integer values of results is suggested. To be perfectly correct two significant figures of uncertainty if the first digit is 1,2 or 3 should be presented but here such accuracy is not necessary. The manuscript should be improved with regards to English usage.

Author Response

The manuscript is focused on the identification of protein complex from sea cucumber (Eupentacta fraudatrix). The complex of MW 2 MDa was isolated by GPC and further analyzed by gel electrophoresis and MALDI. In total 15 proteins and 21 peptides were identified with those methods. A shortcoming of this manuscript is poor characterization of the constituents and qualitative analysis of the complex. In the case of proteins of MW >10 kDa only SDS-PAGE results, while for peptides only MALDI results were presented. No other quantitative or qualitative experiments were attempted to characterize the constituents of the complex. This part of the manuscript should be extended. As a minor remarks the presentations of uncertainties should be corrected. The authors indicated single or double digit uncertainty while corresponding results were presented as if they were measured with lower or higher accuracy. For example 55.2+/-3 or 24 +/- 2.0% (section 2.2 and 2.3). Instead, single digit uncertainty and only integer values of results is suggested. To be perfectly correct two significant figures of uncertainty if the first digit is 1,2 or 3 should be presented but here such accuracy is not necessary. The manuscript should be improved with regards to English usage.

Answer:

Sorry, but this is our first article on such an unusual marine organism. It is not entirely clear to us what you mean by pointing to “no other quantitative or qualitative experiments were attempted to characterize the constituents of the complex. At the moment, it is not yet possible to carry out a more detailed analysis for the identification of proteins with a MW >10 kDa because of the absence of recuired protein databases. As for a more detailed identification of proteins in the future, at present, one of the institutes in Russia is working on the creation of a database of components of sea cucumber. After the publication of this database, it will be possible to refine proteins of the complex. At the moment, their primary characterization can be carried out by molecular weights after SDA-PAGE.  

As for the peptides, as follows from our analyzes of peptides from stable complexes from milk, placenta and sea urchins, they do not correspond to tryptic hydrolysates of any large proteins with mol masses > 10 kDa. Therefore, the establishment of their structure and possible biological role in the composition of the complexes is a long-term and very difficult task that requires additional development of methods of analysis. We plan to solve this problem in the future, and at present we are searching for adequate methods for analyzing the structure of such peptides in a microvariant.

Because of the temporary absence of the database, we plan in the near future to try to identify possible enzymes of the stable complex by analyzing their enzymatic function. However, this is also a time-consuming job. This organism attracts our high attention because it is endowed with the ability to completely re-generate organs and tissues. Therefore, it is a valid model organism for studying the processes of re-generation of organs and tissues. Therefore, we intend, using different approaches, to carry out a complete analysis of the peptides, proteins and enzymes that make up the stable complex over time.

Inaccuracies in the indication of numbers have been completely removed.

Hopefully for the first report taking into account indicated difficulties more detailed study,  the article meets the requirements of the journal

All changes in the text are indicated in red.

Thank you very much for the very helpful comments. This is our first article on such an unusual marine organism. We will continue to explore it in more detail.

Reviewer 2 Report

The manuscript “Very stable two mega Dalton high-molecular-mass multipro-tein complex from sea cucumber Eupentacta fraudatrix” by Anna M. Timofeeva and co-Authors describes a study focused on the isolation and characterization of a high molecular mass protein complex from sea cucumber. This organism attracts high attention because it is endowed with the ability to completely re-generate organs and tissues. Therefore, it is a valid model organism for studying the processes of re-generation of organs and tissues.

In my opinion, this manuscript has originality, it is well written and the analysis, interpretation, discussion and presentation of results are competently performed. The used methodologies are appropriate. I have noted only a few language mistakes in text, which need correction.

It is a pity that the mass-spectrometry results did not provide indications on the identity and function of the proteins contained in the complex, but this is an issue independent of the Authors because it is due to the insufficient entries in the protein databases.

Specific comments.

  1. Introduction. “The extremely stable complexes of different proteins (~1000 ± 100 kDa) were isolated from milk, placentas, and eggs of sea urchins…” I would suggest the addition of a few details (protein identity, function) from the literature about the proteins contained in the complexes isolated from milk, placentas and eggs of sea urchins.
  2. Pag 4, Par 2.3. “…it was impossible to identify none of them.” For future investigations, the Authors could take into consideration the possibility to collect data by direct protein sequencing (using EDMAN degradation chemistry) of isolated proteins. These data could be used to obtain DNA probes to be exploited for gene or cDNA sequencing to obtain sequences of the proteins contained in the complex isolated from sea cucumber.
  3. Pag 4, Par 2.3. “This is due to the absence of data on such cucumber proteins in the databases of sea invertebrates.” It is expected that the Authors already searched a database containing the proteins of sea urchin eggs.
  4. Pag 7, Par 4.3. “The concentrated homogenate (0.5 - 1.0 ml) were applied on Sepharose 4B a column with (volume 50 ml) equilibrated in TBS…” Should it be “The concentrated homogenate (0.5 - 1.0 ml) was applied on a Sepharose 4B column (volume 50 ml) equilibrated in TBS…”?
  5. Pag 4, Par 2.2. “This may indicate in favor of the fact…” I suggest to change this text with “This suggest….”
  6. Pag 4, Par 2.2. “…0.1 EDTA…” Should it be “…0.1 M EDTA…”?
  7. Pag 8, Conclusions. “…time reveal a very…” Should it be “…time revealed a very…”?

Author Response

The manuscript “Very stable two mega Dalton high-molecular-mass multipro-tein complex from sea cucumber Eupentacta fraudatrix” by Anna M. Timofeeva and co-Authors describes a study focused on the isolation and characterization of a high molecular mass protein complex from sea cucumber. This organism attracts high attention because it is endowed with the ability to completely re-generate organs and tissues. Therefore, it is a valid model organism for studying the processes of re-generation of organs and tissues.

In my opinion, this manuscript has originality, it is well written and the analysis, interpretation, discussion and presentation of results are competently performed. The used methodologies are appropriate. I have noted only a few language mistakes in text, which need correction.

It is a pity that the mass-spectrometry results did not provide indications on the identity and function of the proteins contained in the complex, but this is an issue independent of the Authors because it is due to the insufficient entries in the protein databases.

Specific comments.

  1. Introduction. “The extremely stable complexes of different proteins (~1000 ± 100 kDa) were isolated from milk, placentas, and eggs of sea urchins…” I would suggest the addition of a few details (protein identity, function) from the literature about the proteins contained in the complexes isolated from milk, placentas and eggs of sea urchins.

Answer:

We have added information about proteins of three different stable complxes.

It was shown that placenta SPCs contain twelve proteins: hemoglobin, alkaline phosphatase, cytoplasmic actin, human serum albumin, chorionic somatomammotropin hormone, heat shock protein beta-1, peroxiredoxin-1, 78 kDa glucose-regulated protein, protein disulfide isomerase A3, serotransferrin, annexin A5, and IgGs [19,11]. Human milk SPCs contain α-lactalbumin and lactoferrin as major proteins, whereas human milk albumin, alkaline phosphatase, β-casein, sIgAs, and IgGs were present in moderate amounts [9]. Identification of sea urchin complex proteins was performed using several different databases including of sea invertebrates [12]. Major proteins of sea urchin complex had homology with keratin (type II cytoskeletal 1; Homo sapiens), alkaline phosphatase of Pseudomonas fluorescens, human lactotransferrin, glucose-6-phosphatase of Strongylocentrotus purpuratus, and one protein of Alveinella pompejana (cDNA clone CAGA18424 5’, mRNA sequence). Several remaining major and average proteins were impossible to identify even thru search for their possible homology with proteins of different other organisms [12]. The only common protein in all three stable complexes was alkaline phosphatase. This suggests that stable complexes of different protein compositions can also exist in other organisms and have a specific biological role.

It should be noted that in the case of the sea urchin complex, we were unable to identify all proteins due to the lack of an extended database for sea urchins and other sea invertebrates. And yet, in the case of the sea urchin complex, it was possible to find the homology of some proteins with those in mammals and other marine organisms. For sea cucumber, the situation turned out to be more complicated - we have not yet found the homology of cucumber proteins with proteins of other organisms

  1. Pag 4, Par 2.3. “…it was impossible to identify none of them.” For future investigations, the Authors could take into consideration the possibility to collect data by direct protein sequencing (using EDMAN degradation chemistry) of isolated proteins. These data could be used to obtain DNA probes to be exploited for gene or cDNA sequencing to obtain sequences of the proteins contained in the complex isolated from sea cucumber.

Answer:

It is possible that some proteins and enzymes of sea cucumber have weak homology with proteins that have the same function in other organisms. Taking this into account, we plan in the near future to try to identify possible enzymes of the complex by analyzing their enzymatic function. As for a more detailed identification of proteins in the future, at present, one of the institutes in Russia is working on the creation of a database of components of sea cucumber. After the publication of this database, it will be possible to refine proteins without enzymatic function.

  1. Pag 4, Par 2.3. “This is due to the absence of data on such cucumber proteins in the databases of sea invertebrates.” It is expected that the Authors already searched a database containing the proteins of sea urchin eggs.

With the database for the sea urchin, the situation is almost the same as for the sea cucumber. However, while studying the proteins of the Mogo cucumber complex, we still managed to identify some proteins and using databases for other organisms.

Identification of sea urchin complex proteins was performed using several different databases including of sea invertebrates [12]. Major proteins of sea urchin complex had homology with keratin (type II cytoskeletal 1; Homo sapiens), alkaline phosphatase of Pseudomonas fluorescens, human lactotransferrin, glucose-6-phosphatase of Strongylocentrotus purpuratus, and one protein of Alveinella pompejana (cDNA clone CAGA18424 5’, mRNA sequence). Several remaining major and average proteins were impossible to identify even thru search for their possible homology with proteins of different other organisms [12].

  1. Pag 7, Par 4.3. “The concentrated homogenate (0.5 - 1.0 ml) were applied on Sepharose 4B a column with (volume 50 ml) equilibrated in TBS…” Should it be “The concentrated homogenate (0.5 - 1.0 ml) was applied on a Sepharose 4B column (volume 50 ml) equilibrated in TBS…”?

Answer: It was corrected

  1. Pag 4, Par 2.2. “This may indicate in favor of the fact…” I suggest to change this text with “This suggest….”

Answer: It was corrected

  1. Pag 4, Par 2.2. “…0.1 EDTA…” Should it be “…0.1 M EDTA…”?

Answer: It was corrected

  1. Pag 8, Conclusions. “…time reveal a very…” Should it be “…time revealed a very…”?

Answer: It was corrected

All changes in the text are indicated in red

Thank you very much for the very helpful comments. This is our first article on such an unusual marine organism. We will continue to explore it in more detail.

Sincerely

Prof. Georgy A. Nevinsky